# Evaluation of Two Predictive Models for Forecasting Olive Leaf Spot in Northern Greece

**DOI:** 10.3390/plants10061200

**Published:** 2021-06-12

**Authors:** Thomas Thomidis, Konstantinos Michos, Fotis Chatzipapadopoulos, Amalia Tampaki

**Affiliations:** 1Department of Nutritional Science and Diabetics, International Hellenic University, Sindos, 57400 Thessaloniki, Greece; 2Neuropublic S.A., Information Technologies & Smart Farming Services, Piraeus, 18545 Attica, Greece; k_michos@neuropublic.gr (K.M.); f_chatzipapadopoulos@neuropublic.gr (F.C.); a_tampaki@neuropublic.gr (A.T.)

**Keywords:** leaf wetness, temperatures, validation, *Venturia oleaginea*

## Abstract

Olive leaf spot (*Venturia oleaginea*) is a very important disease in olive trees worldwide. The introduction of predictive models for forecasting the appearance of a disease can lead to improved disease management. One of the aims of this study was to investigate the effect of temperature and leaf wetness on conidial germination of local isolates of *V. oleaginea*. The results showed that a temperature range of 5 to 25 °C was appropriate for conidial germination, with 20 °C being the optimum. It was also found that at least 12 h of leaf wetness was required to start the germination of *V. oleaginea* conidia at the optimum temperature. The second aim of this study was to validate the above generic model and a polynomial model for forecasting olive leaf spot disease under the field conditions of Potidea Chalkidiki, Northern Greece. The results showed that both models correctly predicted infection periods. However, there were differences in the severity of the infection, as demonstrated by the goodness-of-fit for the data collected on leaves of olive trees in 2016, 2017 and 2018. Specifically, the generic model predicted lower severity, which fits well with the incidence of the disease symptoms on unsprayed trees. In contrast, the polynomial model predicted high severity levels of infection, but these did not fit well with the incidence of disease symptoms.

## 1. Introduction

Olive leaf spot (*Venturia oleaginea* (Castagne) Rossman & Crous, comb. nov.) is the cause of a very important disease in olive trees worldwide. According to Trapero and Blanco [1] and Viruega et al. [2], *V. oleaginea* over summers as mycelium in infected leaves that remain in the tree canopy or fallen to the soil surface, while in autumn, mycelia resume growth from the latent infections caused during the last spring or from old lesions, and new conidia are produced, which are dispersed by rain splash and run-off. The main symptoms of the disease are dark sooty spots (commonly known as peacock spots) which appear on the upper surface of leaves, mainly in the low canopy. Rarely, similar spots may also appear on the stem and fruit [3]. Heavy premature defoliation, which sometimes leads to twig death of olive (*Olea europaea* L.), can been caused by this pathogen [4] when no preventive or curative sprays are applied. According to Prota [5], in the Mediterranean region, fungicides are usually applied in the main shoot-growth seasons (spring and/or autumn).

Meteorological factors play a key role in infection by the olive leaf spot fungus. Temperature and moisture are the main climate factors influencing the development of *V. oleaginea* in olive trees. Relatively mild to low temperatures and free moisture on the leaves favor infections during the rainy periods in fall, winter, and spring [6,7,8]. Previous works have shown that the minimum temperature for conidia germination of the fungus is 5 °C, the optimum 20 °C, and the maximum 30 °C [6,9,10]; this pathogen is able to sporulate at temperatures from 5 to 25 °C [11]. In Greece, these temperatures occur mainly in the period of September to May [12]. Saad and Masri [13] also demonstrated the relationship between conidial germination and leaf wetness duration. They found that a minimum of 42 h leaf wetness was required for *V. oleaginea* conidia to germinate at 12 °C, while at 20 °C, 18 h was required. Infection occurred from 5 to 25 °C, and disease severity was the greatest at ~20 °C for wetness durations of 12 to 24 h and at ~15 °C for longer durations, while the optimum temperature and minimum wetness durations for infection were 15.5 °C and 11.9 h [6,12,14]. Obanor et al. [15] found that temperature affected olive leaf spot severity, with the lesion numbers increasing gradually from 5 °C to a maximum at 15 °C, and then declining to a minimum at 25 °C, while the numbers of lesions increased with increasing leaf wetness period at all temperatures tested. The minimum leaf wetness periods for infection at 5, 10, 15, 20 and 25 °C were 18, 12, 12, 12 and 24 h, respectively.

Several forecasting models to predict the infection for specific plant diseases have been developed. Each of the developed models has their strengths and weakness, so choosing the right one is based on many factors. A widely used generic infection model is that developed by Magarey et al. [16], which described pathosystems in which the parameters of temperature and wetness duration were supplied for each of the studied pathogens. In addition to the generic model, an infection model using regression equations, such as those based on polynomials, logistic equations can be developed by conducting combinations of multiple temperature and wetness. A forecasting model to predict olive leaf spot infection was developed by Obanor et al. [15] based on a polynomial equation with linear and quadratic terms of temperature, wetness and leaf age. However, this model has not been validated under field conditions.

The successful development of a plant disease forecasting system also requires the proper validation of a developed model. There are a large number of predictive models for the many important plant diseases in the international [17] literature. However, the accuracy of the predictions of each model must be tested under field conditions in order to reduce: (a) erroneous indications of high risk in cases where, in fact, no disease was observed, and (b) erroneous indications of no risk where, in fact, the disease was observed [18]. Although the effect of temperature and leaf wetness on the conidial germination of *V. oleaginea* has been previous studied [15], repetition to validate previous results with local isolates of the pathogen is essential to fit these parameters in the generic model developed by Magarey et al. [16] for local uses. Thus, one of the main aims of this study was to investigate the minimum, maximum and optimum temperatures and leaf wetness durations for conidia germination of local isolates of *V. oleaginea*. Because validation of prediction models for forecasting plant disease under field conditions is important prior to commercial use, a second aim of this study was to validate the above generic model and the polynomial model developed by Obanor et al. [15] under the field conditions in Potidea Chalkidiki, Northern Greece.

## 2. Results

### 2.1. Effect of Temperature and Leaf Wetness on Conidial Germination

There was no significant difference between repeated trials (*p* = 0.201), so the data from the two trials were pooled. Temperature significantly influenced (*p* < 0.001, SE = 0.687) conidial germination. Under continuous wetness, the optimum temperature for conidial germination was 20 °C, whereas conidial germination was inhibited at 30 and 0 °C. Conidial germination was significantly less at 15 and 25 °C than at 20 °C. The percentage of conidial germination was significant higher at 15 and 25 °C than at 10 °C. Conidial germination was significantly less at 5 than 10 °C. The estimates of the parameters from the quadratic function of temperature (R^2^ = 0.739; Y = 7.46 + 5.22 × X − 0.16 × X^2^) and leaf wetness (R^2^ = 0.946; Y = 23.88 + 4.43 × X − 0.06 × X^2^) are presented in Figure 1.

There was no significant difference between repeated trials (*p* = 0.186), so the data from the two trials were pooled. Leaf wetness also significantly influenced (*p* < 0.001, SE = 0.073) conidial germination. Under constant temperature at 20 °C, the conidial germination started after 12 h of continues wetness. In contrast, no conidial germination was observed after 6 h of wetness. The percentage of conidial germination after 18 h of wetness was significantly higher than 12 h, but significantly less than 24 h. No significant difference was observed in the percentage of conidial germination after 24, 36 and 48 h of leaf wetness. The estimates of the parameters from the quadratic function are presented in Figure 2.

### 2.2. Evaluation of Model Accuracy

The average temperature, rainfall, and leaf wetness for the period May to December for each of these three years is presented in Figure 3. Figure 4 presents the predictions of the generic and polynomial models for the period of May to December for three consecutive years (2016, 2017, 2018).

In 2016, the polynomial model predicted risk >29 on the 15th, 20th and 31st of May, 23th September, 10th and 15 October, 8th November and 1st December (Table 1). In contrast, no prediction of risk >29 was given from the generic model in the period April–December 2016. Very low incidence symptoms (percentage of diseased leaves < 5%) of the disease were observed in the unsprayed control trees at the same period. The mean temperature was 18.6 °C in May, which increased to 24.8 °C, 26.6 °C and 26.2 °C in June, July and August, and decreased to 22.7 °C, 17.8 °C, 13.8 °C in September, October and November respectively. The above temperatures were not a limiting factor for infections of the olive trees by the fungus *V. oleaginea*. In contrast, the total degree of hourly leaf wetness was very low in the period of May to December and not favorable for the development of the disease.

In 2017, the polynomial model predicted nearly continuously risk >29 between the 5th and 31st of May. The generic model predicted risk >29 at 15th and 31st May (Table 1). The first symptoms were observed at 28th May, and the final incidence of the symptoms was moderate when recorded 15 days later (percentage of diseased leaves in unsprayed trees was 18%). The mean temperature was 19.8 °C in May, while the total degree of hourly leaf wetness per day was about 20 in the same period. Those climate conditions were favorable for infections of olive trees by the fungus *V. oleaginea*. The optimum temperature and leaf wetness for growth of *V. oleaginea* occurred from 15th to 28th May, justifying the short incubating period of 14 days.

In 2018, the polynomial model predicted risk >29 at 2–5th, 10th, 20th, 25th and 30th May, 8th and 30th June, 25 July, 28th September (Table 1), while the predicted risk >29 was nearly always the period October-December. In contrast, no prediction of risk >29 was given from the generic model in the period April–September 2018. The generic model predicted risk >29 at the 3rd, 6th, and 20th October, 26th October, 16th November, 22th and 28th November, 3rd, 8th and 15th December. The first symptoms of the disease were observed at the 26th October, while the incidence of the symptoms was relatively high at the 10th November (percentage of diseased leaves <36%). No other results were collected after 10 November. The mean temperature was 21.7 °C in May, which increased to 24.6 °C, 26.6 °C and 27.4 °C in June, July and August, and decreased to 23.4 °C, 18.8 °C, 14.3 °C, 9.2 °C in September, October, November, December, respectively. The above temperatures were not a limiting factor for infections of the olive trees by the fungus *V. oleaginea*. The total degree of hourly leaf wetness was very low in the period of May to September, and not favorable for the development of the disease. In contrast, there was a high number of hourly leaf wetness in October, making the climate conditions favorable for the development of the disease.

The estimates of the parameters from the linear regression analysis to find the relationship between model predictions and level of the disease (Generic Model: R^2^ = 0.917, Y = 4.74 + 0.47 × X, Beta Value = 0.958; Polynomial Model: R^2^ = 0.578, Y = 7.86 + 0.34 × X, Beta Value = 0.76) are presented in Figure 5.

## 3. Discussion

So far, control of olive leaf spot has been based on prognosis. This method is adequate, but possesses disadvantages including inopportune and unnecessary spray applications. It increases the cost of production, and also the risk of environmental pollution. The introduction of predictive models to forecast the appearance of a disease could improve crop management by reducing the number of spray applications and improving the effectiveness of spray applications conducted.

Magarey et al. [16] developed a generic model appropriate for predicting the appearance of a high number of plant diseases. This model requires some climate parameters such as the minimum, maximum and optimum temperatures, as wells as the minimum and maximum numbers of hours of leaf wetness. The results of this study showed that a temperature range of 5 to 25 °C was appropriate for the conidial germination on detached olive leaves with 20 °C being the optimum. Previous works showed that conidial germination of *S. oleagina* (synonym of *V. oleaginea*) on agar was at its minimum at 5 °C, with an optimum at 20 °C, and a maximum at 30 °C [9,19,20], while Saad and Masri [13] found that conidial germination of *S. oleagina* could be observed in temperatures ranging from 5 to 25 °C. The above range of temperatures at which *V. oleaginea* conidia germinate suggests that infection may occur mainly throughout the period of September–June in olive growing regions of Northern Greece. This study also showed that at least 12 h of leaf wetness was required to start the conidial germination of *V. oleaginea* at optimum temperatures. According to Obanor et al. [15], the minimum leaf wetness periods for infection of olive trees from *S. oleagina* were 18, 12, 12, 12 and 24 h at 5, 10, 15, 20 and 25 °C, respectively, while the minimum leaf wetness periods required for conidial germination at 5, 10, 15, 20 and 25 °C were 24, 12, 9, 9 and 12 h, respectively [6].

Specific factors, such as pathogen biology, host phenology, and host variety in a specific area may significantly affect the input variables for a predictive model. As the predictive model could contain assumptions about site specific conditions, each model must be validated for a specific location by testing for one or more seasons under local conditions to verify that it works with precision in this location. Obanor et al. [15] developed a regression model to predict the infections of olive trees by the fungus *S*. *oleagina.* However, this model was not evaluated and validated under field conditions. In this study, the generic model developed by Magarey et al. [16] and the polynomial model developed by Obanor et al. [15] were evaluated to predict infection of olive trees by the fungus *V. oleaginea* under the climate conditions of Potidea Chalkidiki, Northern Greece. Because the purpose of the model was to be part of a warning system for olive leaf spot management, the ability to correctly predict infection periods is crucial. The results showed that both models correctly predicted infection periods. However, there was difference in the severity of the infection, as demonstrated by the goodness-of-fit for the data collected on leaves of olive trees in 2016, 2017 and 2018. Specifically, the generic model predicted lower severity of the infection which fits very well with the incidence of the symptoms of the disease. In contrast, the model developed by Obanor et al. [15] predicted high severity of the infection, but these did not fit as well with the incidence of the symptoms. Based on the above results, the polynomial model gave false positive predictions and did not generate proper spray recommendations increasing the fungicide applications with indirect results the increase of cost production and possible environmental pollution. It is recommended that the polynomial model be calibrated and re-validated under field conditions before commercial use. In contrast, the generic model gave a correct prediction for the appearance of the disease, and it seems to fit better in computer-assisted Decision Support Systems (DSSs).

Considering that this study did not include olive cultivars with different levels of susceptibility, it was not possible to evaluate whether each of the above predictive model can be fitted better to specific olive cultivars as the same climate conditions could favor different incidence of the symptoms depending on the level of cultivar susceptibility.

## 4. Materials and Methods

### 4.1. Effect of Temperature and Leaf Wetness on Conidial Germination

#### 4.1.1. Effect of Temperature

The effect of temperature on conidial germination of *V. oleaginea* was investigated by using the methodology described by Obanor et al. [6]. Leaves (cv. Chondrolia Chalkidikis) with symptoms of olive leaf spot were collected from a commercial field established in Potidea Chalkidiki (40.1939° N, 23.3301° E) in November 2015. The leaves were agitated in distilled water and the conidial suspension filtered through a double layer of cheesecloth to remove leaf debris. Inoculum suspensions were adjusted to 6 × 10^4^ conidia mL^−1^ by using a hemocytometer. Seven temperatures (0, 5, 10, 15, 20, 25, and 30 °C) were tested to find the upper and lower limits of spore germination. Fully expanded leaves (4 weeks old) without symptoms of the disease were excised from olive trees grown in a commercial field by cutting at the stem end of the petiole. The leaves, before being inoculated, were disinfested by dipping them in 10% vol/vol domestic bleach solution (4.85% NaOCl) for 5 min, washed three times with sterile distilled water and left to dry at room temperature. The leaves were inoculated with two drops (10 μL) of the conidial suspension deposited on the upper leaf surface. After inoculation, the leaves were placed in 9-cm petri dishes (wet paper towel was placed onto bottom and leaves was placed on plastic sticks so that to avoid any contact) arranged randomly in the growth chamber (97–100% RH) described below. Results were collected by recording the germination of 30 conidia/leaf (10 leaves for each treatment) 24h later. A conidium was considered germinated when the germ tube was equal to the greatest diameter of the swollen conidia (1 to 1.5×) [21,22].

#### 4.1.2. Effect of Leaf Wetness

Similarly, fully expanded leaves of olive trees (cv. Chondrolia Chlakidiki) without symptoms of disease were inoculated with two drops (10 μL) of the conidia suspension as described above. After inoculation, the leaves were placed in 9-cm petri dishes (wet paper towel was placed onto the bottom, and leaves were placed on plastic sticks so as to avoid any contact) arranged randomly in the growth chamber (Emmanuel E. Chryssagis, Growth Plant Chambers—GRW 500/CMP2) (97% ± 3 Relative Humidity) under continuous wetness at the 20 °C (optimal temperature identified above) and incubated for 6, 12, 18, 24, 36 and 48 h. Results were collected by recording the germination of 30 conidia as described above.

Both experiments were repeated once. General linear regression analysis was performed (SPSS Grad Pack 23, SPSS Inc., Chicago, IL, USA) in order to determine the relationship between leaf wetness, temperatures and conidia germination.

### 4.2. Model Development and Validation

#### 4.2.1. The Models

The generic model developed by Magarey et al. [16] was used. The parameters were used to run the model based on the results produced in the above experiments: Minimum Temperature (Tmin) = 5 °C, Maximum Temperature (Tmax) = 25 °C, Optimum Temperature (Topt) = 20 °C, Minimum Leaf Wetness (Wmin) = 12 h, Maximum Leaf Wetness (Wmax) = 24 h. In addition to the above, the predictive model (polynomial model; Y = β0 + β1A + β2Τ + β3W +β4(Ax W) + β5Τ2 + β6W2), where Y is √(disease severity), A is the leaf age (weeks), T is temperature (°C), W is wetness period (h), and β0....β6 are determined parameters)to forecast the appearance of olive leaf spot developed by Obanor et al. [15] was simultaneously evaluated under the field conditions of Potidea Chalkidiki.

The leaf wetness was estimated from the hourly data: if an (i) hour is wet, it is counted as 1, or when it is dry it is designated as 0 (so the dry hours are not counted and are not taken into account). Continuous wet hours are summed to determine leaf wetness. However, if there is an interruption of fewer than or equal to 20 dry hours and low relative humidity (<70%) (based on the result published by Villalta et al. [23] for the fungus *Venturia pirina*), the summation of hours is continued. In contrast, if the interruption of dry hours is longer than 20 dry hours, a new summation of hours is started. Temperature is the event average temperature during each wet period. Cultivar susceptibility and inoculum level were not considered because insufficient information was available about their effects on the occurrence of infection.

#### 4.2.2. Evaluation of Model Accuracy under Field Conditions

Model accuracy in predicting the day of infection was evaluated by comparing actual and predicted times of symptom appearance. In the Potidea Chalkidiki, which is one of the most important olive production areas in Greece, a telemetric meteorological station (NEUROPUBLIC S. A., Information Technologies & Smart Farming Services, Piraeus, 18545, Attica, Greece) was established to record weather data, which were used to run the models. The model was operated hourly, starting from the 1st of May (aiming to include both periods favorable for the development of the disease (May to June) and unfavorable periods for the development of the disease (July to August)), 00.01, and ending at 31st December using hourly leaf wetness and hourly temperatures as driving variables for calculation. The date of the first observation of the symptoms (in young leaves) was used to verify the prediction of the models, while the final incidence of the symptoms was recorded 15–20 days later by calculating the percentage of diseased leaves to a sample of 100 leaves randomly selected from each of 10 trees in total. The period of possible appearance of the disease was calculated on each day when Risk (LW, T) > 30, which was considered to be the incubation period. According to Bakarić [24], an incubation period depends on the environmental conditions, and it lasts 15 days, but can be extended from three to eight or more months. The model predictions were ranged from 0 (when Risk = 0) to 100 (when Risk = the highest possible value).It was calculated with 0 being the minimum value that could be given by the model, and 100 the maximum value. All the other values were distributed between 0 and 100. Previous preliminary work under field conditions to find the threshold for the model predictions showed that no symptoms or very light-sporadic symptoms (percentage of diseased leaves < 5%) of the disease could be observed when the model predictions were in the range 0–29 (indicating that a spray application against olive leaf spot disease would not be financially justifiable; the spray program usually includes copper-based fungicides applied before the onset of the main infection periods, which often coincide with the main shoot-growth seasons (spring and autumn)).

A commercial olive field (cv. Chondrolia Chalkidiki, 7- to 10-year-old trees), located in Potidea, Chalkidiki, was chosen to record the appearance of olive leaf spot symptoms. Trees were pruned to a vase shape by hand pruning. Five to six irrigations were provided yearly. Nitrogen was applied yearly as (NH_4_)_2_SO_4_ at 100 N units per hectare. Selected trees did not show any symptom of the disease before starting the trial. The trees (kept unsprayed) were inspected twice per week to determine the time of symptom onset. The trees were carefully inspected for the appearance of the first symptoms, which are dark sooty spots (commonly known as peacock spots) appear on the upper surface of leaves, mainly in the low canopy. When the symptoms were unclear, the leaves were marked and observed during the following surveys. To assess the severity of the disease, 100 random leaves were observed for the symptoms of the disease per tree (results were collected from 10 trees), and the disease incidence was calculated as the percentage of leaves with leaf spot symptoms. The predicted period of disease onset was then compared with the actual one. The model was judged to have provided an accurate prediction when the observed symptom onset coincided with the time interval predicted by the model [25].

This experiment was conducted for three consecutive years (2016, 2017, and 2018). General linear regression analysis was performed (SPSSGradPack23, SPSS Inc., Chicago, IL, USA) in order to determine the relationship between model predictions and level of the disease (observations).

## 5. Conclusions

The effects of the air temperature and leaf wetness on *V. oleaginea* infection of olive leaves were clarified. Based on those results, disease forecasting systems were developed to find the proper timing of foliar fungicidal sprays. The generic model predicted lower severity, which fits well with the incidence of the symptoms of the disease in unsprayed trees, and this model could be used to schedule the spray applications against olive spot disease. In contrast, the polynomial model predicted high severity levels of infection, but this did not fit well with the incidence of the symptoms. This study could help pest managers and researchers predict the risk of olive leaf spot in different regions or under different crop management practices.

## Figures and Tables

**Figure 1 plants-10-01200-f001:**
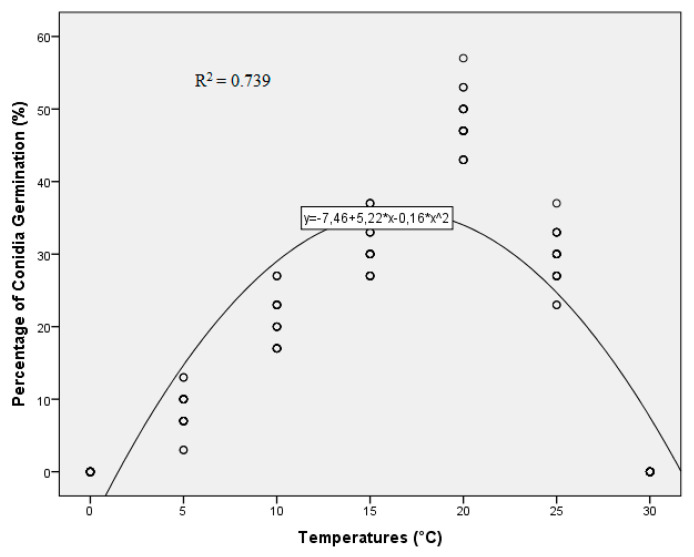
Effect of temperature on conidial germination of *Venturia oleaginea.* The parameters of minimum, maximum and optimum temperatures were fit to the generic model.

**Figure 2 plants-10-01200-f002:**
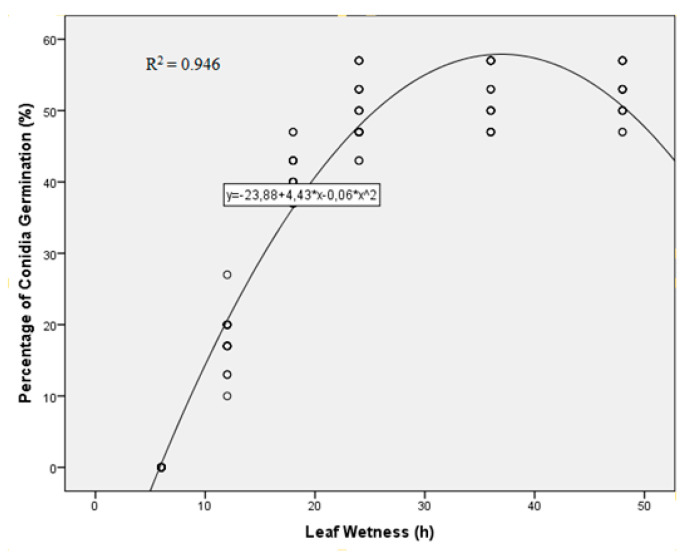
Effect of leaf wetness on the conidia germination of *Venturia oleaginea.* The parameters of minimum and maximum leaf wetness were fit to the generic model.

**Figure 3 plants-10-01200-f003:**
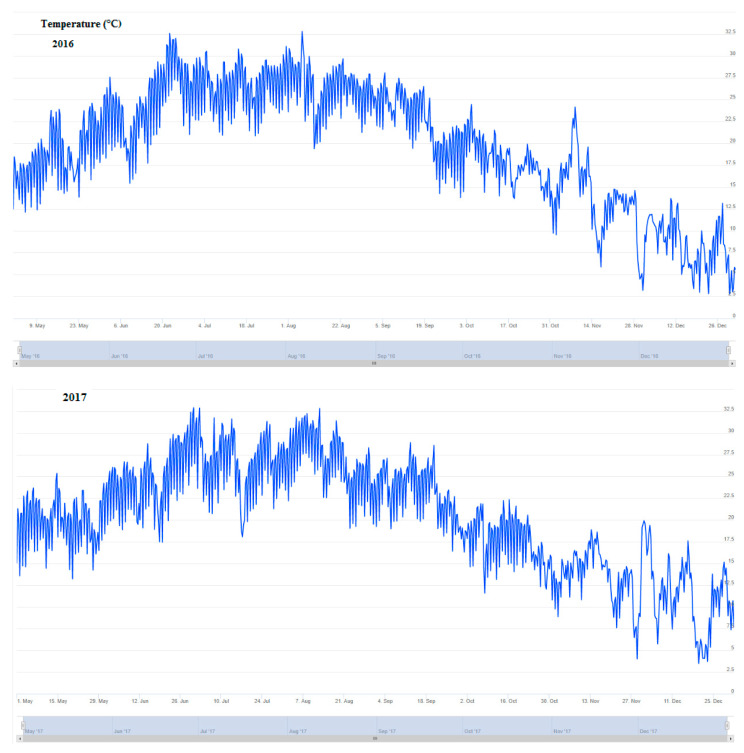
Average temperature, rainfall, and leaf wetness for the period May to December in 2016, 2017 and 2018.

**Figure 4 plants-10-01200-f004:**
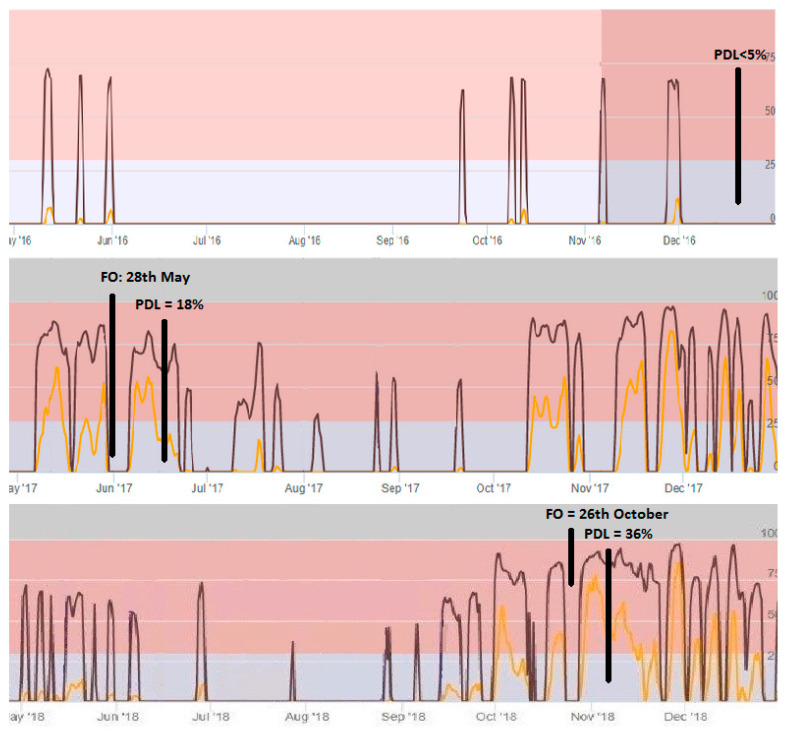
Predictions (as presented on the screen of the computer) of the generic model (orange line) and polynomial models (black line) to forecast infections from the fungus *Venturia oleaginea* on olive trees for the period May to December of three consecutive years (2016, 2017, 2018) (FO = First Observation; PDL = Percentage of the Diseased Leaves).

**Figure 5 plants-10-01200-f005:**
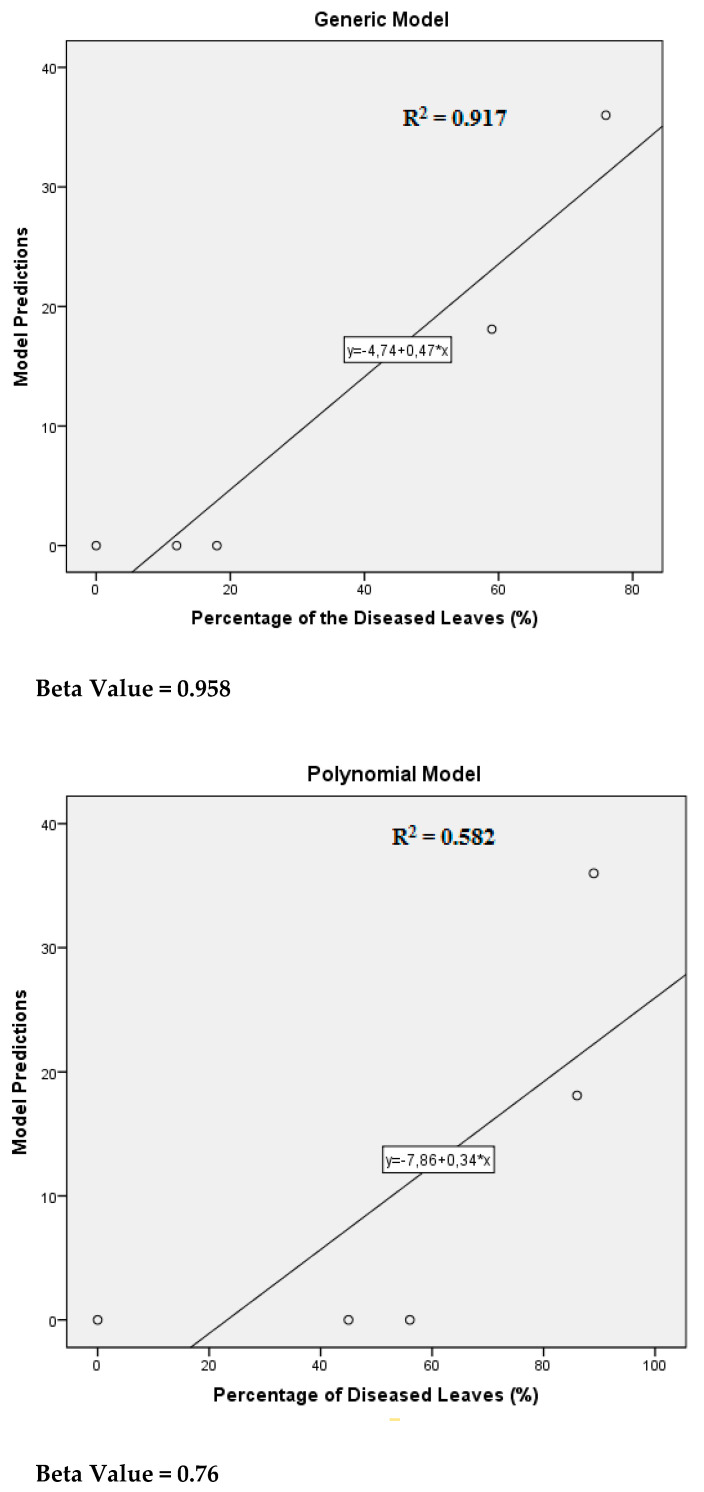
Relationship between model (generic and polynomial) predictions and percentage of diseased leaves.

**Table 1 plants-10-01200-t001:** Dates of the first seasonal infection of *Venturia oleaginea* predicted by the Generic and Polynomial models in 2016, 2017 and 2018; corresponding values of the risk index calculated by the model; times of actual disease onset; and percentage of diseased leaves.

Year	Date of First Seasonal Infection Predicted by Model (>29) ^a^	Risk Value ^b^	Time (Days) of Actual Disease Onset ^c^	Percentage of Diseased Leaves
Generic Model				
2016	-	-	-	<5%
2017	15th May	59	14	18%
2018	3rd October	66	23	36%
Polynomial Model				
2016	15th May ^d^	56	-	<5%
2017	5th May ^d^	86	24	18%
2018	2rd October ^d^	64	25	36%

^a^ Date when model prediction value is higher to 29. ^b^ Risk was calculated on a scale from 0 (no risk) to 100 (maximum risk) based on weather data and tree growth stage. ^c^ Days after predicted infection–incubation period. ^d^ Model predicted risk >29 was also observed on other dates of the year without correlated with actual disease onset (see Figure 4).

## Data Availability

The data presented in this study are available on request from the corresponding author. The data are not publicly available due to the agreement with the NEUROPUBLIC S. A.

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
