# Peer review of "Evaluation of Two Predictive Models for Forecasting Olive Leaf Spot in Northern Greece"

_plants, 2021, doi:10.3390/plants10061200_

Round 1

Reviewer 1 Report

  1. Venturia oleaginea and V. oleaginea (like all Latin binomials of species) must be written in italics.
  2. The biological cycle of the pathogen should be reported or described.
  3. The terms “severity”, “intensity”and “incidence” of the disease generally express different concepts. It is advisable to unambiguously clarify which parameter is used to determine the level of infection.
  4. Lines 266-268: the leaves, before being inoculated, must be sterilized on the surface, to avoid the possible interference of the epiphytic microorganisms on the germination of the conidia.
  5. The Conclusions paragraph should be expanded to better describe and clarify the obtained results.
  6. There are several typos throughout the article (for example: line 30, canopy1; l. 141, ungus; l. 320, eachof10, etc.) and unclear points (lines 33; 47-48; 316 -320; etc.). Extensive editing of the English language and style is required.

Author Response

  • Venturia oleaginea and V. oleaginea (like all Latin binomials of species) must be written in italics.

Done

  • The biological cycle of the pathogen should be reported or described.

Done

  • The terms “severity”, “intensity”and “incidence” of the disease generally express different concepts. It is advisable to unambiguously clarify which parameter is used to determine the level of infection.

Corrected

  • Lines 266-268: the leaves, before being inoculated, must be sterilized on the surface, to avoid the possible interference of the epiphytic microorganisms on the germination of the conidia.

This information was added

  • The Conclusions paragraph should be expanded to better describe and clarify the obtained results.

Revised

  • There are several typos throughout the article (for example: line 30, canopy1; l. 141, ungus; l. 320, eachof10, etc.) and unclear points (lines 33; 47-48; 316 -320; etc.). Extensive editing of the English language and style is required

The English were revised by Mr John Cullum, Lecturer at Writtle University College, UK.

Reviewer 2 Report

Dear Authors,

I revised the manuscript "Evaluation of Two Predictive Models to Forecast of Olive Leaf spot in Northern Greece" submitted to Plants Journal. The manuscript is interesting. However, I have some concerns, which need to be addressed before considering for final publication.

  1. Line 30. Remove the number "1" at the end of the word "canopy".
  2. Section "2. Results". Place additionally in the text the quadratic and linear equations that are the results of the calculations and are shown in Figures 1, 2, and 5.
  3. Figure 1. Add the designation degrees "o" of Celsius. Remove the word "Quadratic". This follows from the quadratic function shown.
  4. Figure 2. Remove the word "Quadratic". This follows from the quadratic function shown.
  5. Figure 3. The graphs in Figure 3 are not very clear. Improve their quality.
  6. Figure 5. Remove the word "Linear". This follows from the linear function shown. What unit of measure is on the vertical axis?
  7. Line 204. Instead of the word "spay", use the word "spray".
  8. Line 263-264. Present a method for testing temperatures. What devices did you use?
  9. Section "5. Conclusions". In this section, you should present the main results obtained during your research and relate them to your conclusions. The entire chapter should be expanded.
  10. References. Most of the references are quite old. Try to look for their equivalents in more recent releases. Add some references as there are too few.

Author Response

  1. Line 30. Remove the number "1" at the end of the word "canopy".

Removed

  1. Section "2. Results". Place additionally in the text the quadratic and linear equations that are the results of the calculations and are shown in Figures 1, 2, and 5.

Done

  1. Figure 1. Add the designation degrees "o" of Celsius. Remove the word "Quadratic". This follows from the quadratic function shown.

Done

  1. Figure 2. Remove the word "Quadratic". This follows from the quadratic function shown.

Done

  1. Figure 3. The graphs in Figure 3 are not very clear. Improve their quality.

Done

  1. Figure 5. Remove the word "Linear". This follows from the linear function shown.

Done

What unit of measure is on the vertical axis?

There is no unit. The model index is just a number

  1. Line 204. Instead of the word "spay", use the word "spray".

Corrected

  1. Line 263-264. Present a method for testing temperatures. What devices did you use?

It was growth chambers described below (Emmanuel E. Chryssagis, Growth Plant Chambers - GRW 500/CMP2) (97% ± 3 Relative Humidity)

  1. Section "5. Conclusions". In this section, you should present the main results obtained during your research and relate them to your conclusions. The entire chapter should be expanded.

Done

  1. References. Most of the references are quite old. Try to look for their equivalents in more recent releases. Add some references as there are too few.

The reference list was updated. New references were added

Round 2

Reviewer 1 Report

Suggestions and recommended changes have been made properly.